# The Role of Standardized Phase Angle in the Assessment of Nutritional Status and Clinical Outcomes in Cancer Patients: A Systematic Review of the Literature

**DOI:** 10.3390/nu15010050

**Published:** 2022-12-22

**Authors:** Nan Jiang, Jiaxin Zhang, Siming Cheng, Bing Liang

**Affiliations:** School of Nursing, Jilin University, Changchun 130021, China

**Keywords:** bioelectric impedance analysis, cancer, clinical outcome, cut-off value, nutrition assessment, standardized phase angle

## Abstract

Compared with the phase angle (PA), the predictive ability of the standardized phase angle (SPA) in assessing nutritional status and clinical outcomes in cancer patients remains uncertain. This review aimed to assess (1) the relationship between SPA and nutritional status and clinical outcomes (including complications and survival) in cancer patients; (2) the predictive ability of SPA alone and in comparison with the predictive ability of PA; and (3) the cut-off value of SPA in cancer patients. Studies that addressed the relationship of SPA use to nutritional status, complications, and survival in cancer patients were searched and identified from six electronic databases (PubMed, Medline, CINAHL, Embase, Web of Science, and the Cochrane Library). The included studies were considered to meet the following criteria: English studies with original data that reflected the effects of SPA on nutritional status and clinical outcomes (including complications and survival) and reported a cut-off value of SPA in cancer patients aged ≥18. Thirteen studies that included a total of 2787 participants were evaluated. Five studies assessed the relationship between SPA and nutritional status, and four of them reported a positive relationship between SPA and nutritional status in cancer patients, even considering SPA as a predictor. Twelve studies assessed the relationship between SPA and clinical outcomes in cancer patients. Two-thirds of the studies that evaluated complications reported the predictive ability of SPA; 30% of survival studies reported a positive relationship, 40% reported SPA as a predictor, and 30% reported no relationship. The standard cut-off value for SPA has not yet been determined. Data from the selected studies suggest that SPA might be a predictor of nutritional status. Further studies are needed to determine the value of SPA in predicting nutritional status and clinical outcomes in cancer patients.

## 1. Introduction

Cancer is recognized as a major disease worldwide. GLOBOCAN reported that the numbers of new cancer cases worldwide were approximately 18.1 million and 19.3 million in 2018 and 2020, respectively, and projected that in 2040 this number will be 28.4 million [1]. This indicates that the incidence of cancer is increasing year by year. Studies have shown that cancer patients often experience changes in body composition. The most common change is weight loss [2,3,4], and the loss of skeletal muscle causes a decrease in lean tissue [5,6]. At the same time, a variety of changes in adipose tissue have been shown to occur in different types of cancer [7,8,9]. Cancer patients often suffer from malnutrition [10,11,12] and poor clinical outcomes, including complications [13] and worse survival [14]. These factors can prolong hospitalization time [15], increase families’ financial burdens [16], and even cause massive and disastrous expenditures by public health [17,18]. The European Society for Clinical Nutrition and Metabolism (ESPEN) reported that timely assessment of nutritional status can help patients maintain or gain weight and that this helps improve the effects of therapy and the quality of life of cancer patients [19], and reduces the expenses incurred by their families and government agencies. Therefore, early assessment of the nutritional status of cancer patients is vital. Nutritional status is currently assessed using subjective methods such as the Patient-Generated Subjective Global Assessment (PG-SGA) questionnaire [20] and objective methods such as anthropometry and laboratory measurements [21]. However, these methods of assessment are rarely used in clinical practice due to their characteristics of high heterogeneity, low accuracy, long measurement time, and susceptibility to confounding factors [22]. The prediction of clinical outcomes in cancer patients is subjective and inaccurate when judged by clinicians based on clinical experience [23]. Defects in cellular membrane integrity and fluid balance are characteristic of malnutrition [24], and body composition has been well accepted as an independent predictor of clinical outcomes in cancer patients [25]. Thus, it is possible to assess body composition and use the results of this assessment to predict nutritional status and clinical outcomes in cancer patients.

Bioelectric impedance analysis (BIA) is a safe, noninvasive, simple-to-use, and inexpensive method for indirect assessment of body composition [26]. BIA variables, including age, stature, and weight, can be used to assess fat mass (FFM), skeletal muscle mass (SM), and appendicular skeletal mass (ASM) as predictive equations. As a parameter derived from BIA, phase angle (PA) is an indicator of cell membrane health and integrity as well as of hydration and nutritional status, especially in cancer patients [27,28]. An increasing number of scholars have presented evidence that PA is positively related to nutritional status and clinical outcomes in cancer patients [29,30,31]. Low PA was shown to be related to impaired nutritional and functional status, decreased quality of life, and increased morbidity and mortality [32]. In contrast, high PA indicated better nutritional status and prolonged survival [33]. However, there are individual differences in tissue specificity among patients, and PA is therefore often affected by gender, age, body mass index (BMI), and other factors. Standardization of PA using healthy population reference values makes results universal and comparable across studies and different clinical settings [34]. Standardized phase angle (SPA), which is calculated as [(observed PA—mean PA)/standard deviation of PA] and provides a Z score for PA [35], can quantify individual deviations and allows numerical comparisons among patients of different ages, sexes, and BMIs. SPA can enhance the predictive ability of nutritional status on clinical outcomes by comparing patients’ assessed results with the average of the overall specific population [33]. Sukackiene et al. found that SPA is a good predictor of nutritional status and clinical outcomes in kidney transplant patients [36], and a similar predictive ability of SPA has also been demonstrated in older people [37], critically ill patients [38], cancer patients [34], and other groups of individuals. Even a study conducted during the COVID-19 pandemic reported that SPA could be used to predict survival in patients with COVID-19 [39].

In particular, the results of clinical studies of the effects of using SPA in cancer patients suggest that SPA might be useful for assessing nutritional status and clinical outcomes in cancer patients [13,40]. However, the current evidence is heterogeneous and controversial. In this context, we aimed to more comprehensively assess the relationship between SPA and nutritional status and SPA and clinical outcomes in cancer patients and to define a possible reference cut-off SPA value. The results of our study provide clinicians with information that can be used to assess the applicability of SPA as a potential comprehensive predictive tool for cancer patients.

## 2. Materials and Methods

### 2.1. Study Search

A systematic review was conducted according to the Joanna Briggs Institute methodology for systematic reviews of aetiology and risk evidence [41]. Reporting was performed according to the Preferred Reporting Items for Systematic Reviews and Meta-Analyses (PRISMA) criteria [42]. Two authors conducted a comprehensive search of English studies in several electronic databases (including PubMed, Medline, CINAHL, Embase, Web of Science, and the Cochrane Library) from the establishment of each database until 17 April 2022. The following essential words and their synonyms were used: standardized phase angle, phase angle, and cancer. We also included studies identified through searching by hand. The search strategy is shown in Table 1 and Appendix A.

### 2.2. Study Selection and Selection Criteria

The present study was designed and conducted according to PRISMA statements [42]. This review has been registered in the international prospective register of systematic reviews (PROSPERO) under registration number CRD42022327591. After obtaining the electronic database search results, two authors examined the titles, abstracts, and full texts of the studies. These two authors resolved disagreements at any review stage through negotiation and, if necessary, sought advice from a third person. Each original study was only included once. All types of studies were eligible, and there were no restrictions on sample size, sex, type of cancer, country, or region. Information on the BIA measurements was also not restricted by factors such as machine model or frequency.

The inclusion criteria were as follows: (1) the study participants were patients age ≥18 with cancer; (2) the study reported research on the effects of SPA on nutritional status and clinical outcomes (including complications and survival) in cancer patients; (3) a cut-off value for SPA was mentioned; (4) the article was published in English and included original data.

The exclusion criteria were as follows: (1) the participants were not cancer patients or were cancer patients aged ≤18; (2) SPA was not mentioned; (3) no effects of SPA on nutritional status and clinical outcomes (including complications and survival) in cancer patients, such as researching the screening effect of SPA on cancer or researching the effect of other intervention methods on SPA, were reported; (4) the study combined SPA with other indicators; (5) no cut-off SPA value was mentioned; (6) the article was a review, meta-analysis, meeting abstract, retraction, editorial, letter, personal comment, or book chapter, or did not present original data; (7) the article was not published in English.

### 2.3. Data Extraction

Two authors created a table that was used to extract information and extracted the key information in each study. The information that was recorded in the table included the first author, study title, year of publication, study country, main purpose of the research, study type, participants, sample size, cancer location, BIA machine model used, SPA cut-off value, and main findings. The number of participants in each study was based on the number of patients selected for inclusion in the study. The assessment tools were presented in an abbreviated form with the full name of each tool presented at the end of the form.

### 2.4. Quality Assessment

Two authors used the “Quantitative Non-randomized Studies” section of the Mixed Methods Appraisal Tool (MMAT) scale to assess the quality of the included studies. The MMAT is a reliable and valid tool for assessing the overall quality of the various study designs, which included five types: qualitative studies, quantitative randomized controlled trials, quantitative non-randomized studies, qualitative descriptive studies, and mixed method studies [43]. MMAT can be used to assess the quality of empirical studies, which are based on experimental work, and observational studies or simulations [43]. The specific items that appear in the MMAT are shown in Appendix A; “*” means “meets the criteria”, “-” means “does not achieve the criteria”, and “/” means “we cannot find it in the study”. Two authors marked the corresponding symbols in the corresponding table. Finally, we calculated the total scores of the included studies and assessed their quality levels.

## 3. Results

### 3.1. Study Selection and Description of Studies

A flow diagram of the study selection process is presented in Figure 1. Thirteen quantitative studies were included in this systematic review; they included cross-sectional studies (N = 5) [44,45,46,47,48] and cohort studies (N = 8) [13,34,40,49,50,51,52,53] (Figure 2A). The main characteristics of the included cancer patients and the included studies are described in Table 2 and Table 3. All the included studies were published after 2000. We divided the studies into three groups based on the year of publication: 2001 to 2010 (N = 1) [34], 2011 to 2020 (N = 8) [13,44,45,46,47,49,50,52], and 2021 to 2022 (N = 4) [40,48,51,53] (Figure 2B). The included studies were conducted in various countries at different study centres; they included single-centre studies conducted in the United States (N = 2) [46,47], Brazil (N = 5) [13,44,48,50,52], Sweden (N = 1) [45], Spain (N = 1) [51], Italy (N = 1) [40], and Germany (N = 1) [49], and a multi-centre study conducted in Germany (N = 1) [34]. There was also one multi-centre study that was conducted in both Germany and Italy [53] (Figure 2C). The patients in the included studies suffered from various types of cancer, including head and neck cancer (N = 2) [45,51], colorectal cancer (N = 1) [50], abdominal cancer (N = 1) [40], haematological malignancies (N = 1) [49], and acute leukaemia (N = 1) [47], and some of the studies included patients with mixed types of cancer (N = 7) [13,34,46,47,48,52,53] (Figure 2D). All of the participants in the studies were over 18 years of age. Various types of BIA were used in the studies to assess SPA; these are shown in Table 2 and Figure 2E. Most of the included studies measured SPA at 50 kHz; the cut-off values for SPA at 50 kHz are also shown in Table 3.

### 3.2. Quality Assessment

Quality assessments of the included studies are presented in Appendix A. All studies asked clear research questions that could be addressed by collecting data. All of the included studies were assessed using the “Quantitative Non-Randomized Studies” section of the MMAT. All of the participants in the 13 included studies were representative of the target population, all measures of outcomes were appropriate, and all exposures were as expected. One study had incomplete data due to the deaths of some of the cancer patients [47]; we marked this study as “-” in column 3 of Appendix A. In addition, we marked two studies that did not consider possible confounders in their design and analysis [51,52] as “-” in column 4 of Appendix A. In conclusion, 10 of the included studies met 100% of the quality assessment criteria [13,34,40,44,45,46,48,49,50,53], and 3 met 80% of the quality assessment criteria [47,51,52].

### 3.3. Relationship between SPA and Nutrition Status

Five studies mentioned the relationship between SPA and nutritional status in cancer patients [13,34,47,49,53]; these are shown in Table 4 and Appendix A. In these studies, both subjective nutritional indicators (N = 2, 40%) [13,34] and objective nutritional indicators, including muscle function (N = 3, 60%) [13,51,53] and laboratory measurements (N = 2, 40%) [47,51], were mentioned.

Pena et al. [13] assessed the relationship between SPA and nutritional status in 121 cancer patients who were awaiting surgery. They found that patients with SPA < −1.65 had decreased levels of PT-SGA, a parameter that has been listed as a recommended nutritional assessment tool for adults by the Australian Dietitians Association (DAA) [54]. Lower PT-SGA scores indicate worse nutritional status. In addition, arm circumference (MAC) [13,51], muscular midarm circumference (MMA) [13], calf circumference [51], handgrip strength (HGS) [13], triceps skinfold [13], and thigh adipose tissue [51], all of which are related to muscle function and form a part of the PT-SGA, also decreased with lower SPA. Poorer muscle function indicated poorer nutritional status. Similarly, Leon-Idougourram et al. [51] studied 45 patients with head and neck cancer who were undergoing systemic treatment; in that study, 26 patients with SPA < −1.65 had decreased levels of BMI (*p* = 0.04) and increased levels of C-reactive protein (CRP) (*p* = 0.04) and serum interleukin-6 (IL-6) (*p* = 0.007). Yates et al. [47] found a decrease in albumin levels (*p* = 0.014) when SPA < −1.65 in 100 patients with acute leukaemia. Although in recent years, ESPEN no longer recommends the use of serum albumin to identify adult malnutrition, and indicates that the decrease of serum albumin level is more indicative of the development of inflammation than malnutrition [55], ESPEN guidelines continue to recognize that inflammation is an important potential factor that increases the risk of malnutrition [55,56], suggesting that decreased serum albumin is associated with an increased risk of malnutrition in cancer patients. Therefore, these results of included studies directly or indirectly indicated that there was a higher risk of malnutrition in cancer patients who had lower SPA.

Based on the changes in SPA with nutritional status and related indicators, we considered the relationship between SPA and nutritional status and related indicators, and investigated whether SPA can be used as an indicator of nutritional status in cancer patients. As positive indicators of nutritional status, PT-SGA [13,34], HGS [13,53], MAC [13], MMA [13], BMI [53], and albumin (r = 0.20; *p* = 0.10) [47] were positively related to SPA. In contrast, weight loss [53], a negative indicator of nutritional status, was negatively related to SPA in 1084 cancer patients. Norman et al. [34] reported that SPA was an independent predictor of HGS (coefficient B 1.902; 95% CI, 1.321–2.483; *p* < 0.0001) and that SPA over the 5th percentile value had the strongest positive relationship to both moderate and severe malnutrition in 399 cancer patients, indicating that SPA was an independent predictor of nutritional status in cancer patients.

### 3.4. Relationship between SPA and Clinical Outcomes 

Twelve studies mentioned a relationship between SPA and clinical outcomes in cancer patients (Appendix A). The evaluated clinical outcomes included complications (N = 4, 33.3%, Table 5) [13,40,50,52] and survival (N = 9, 75%, Table 6) [13,34,44,45,46,47,48,49,53].

#### 3.4.1. Relationship between SPA and Complications

Four studies evaluated the relationships between SPA and complications in cancer patients; the results of this evaluation are shown in Table 5.

Cancer patients with lower SPA had more complications [13,40,52]. Pena et al. [13] found that patients with SPA < −1.65 had more infectious complications (including operative wound dehiscence, pneumonia, bacteraemia, urinary tract infection, fistula, sepsis, infection and fever > 38 °C, hypotension, and oliguria) than those with SPA > −1.65. Similarly, Harter et al. [52] found that patients with severe postoperative complications (including complications after surgical or radiological interventions or endoscopy, ICU, organ dysfunction, and even death) had a mean SPA value of −0.71 (−1.44; 0.16), while patients without complications had a mean SPA value of 0.41 (−0.16; 1.07), a significant difference (*p* = 0.007). They reported that patients with SPA < −1.65 had more complications than other cancer patients (*p* = 0.007). Roccamatisi et al. [40] found that SPA < 0.3 was associated with more complications (including multisite infections, infection with multidrug-resistant organisms, and candida coinfection) in patients with abdominal cancer (*p* = 0.032).

Regarding the relevance and predictive value of SPA in cancer patients, Pena et al. [13] reported a negative relationship between SPA and infectious complications in cancer patients (OR 4.19; 95% CI, 1.52–11.53; *p* = 0.006). SPA was considered as an independent predictor of infectious complications [13,40]. However, Pena et al. [13] also reported that there was no relationship between SPA and complications other than infectious ones (OR 1.07; 95% CI, 0.49–2.75; *p* = 0.881), and Maurício et al. [50] found that there was no significant relationship between SPA and postoperative complications in 84 cancer patients (RR 1.53; 95% CI, 0.79–2.92; *p* = 0.199).

#### 3.4.2. Relationship between SPA and Survival

In addition to complications, survival is also an important indicator of clinical outcome in cancer patients. Urbain et al. [49] found that the survival rate of patients with SPA < −2.26 was 37% during a 2-year follow-up, while that of patients with SPA > −2.26 was 51%. In addition to the 2-year survival rate, the 5-year survival rate (*p* = 0.002) [45], the 60-day survival rate (OR 5.25; 95% CI, 1.35–20.44, *p* = 0.02), and the median OS (HR 1.57; 95% CI, 0.93–2.66; *p* = 0.09) [47] of cancer patients with lower SPA were also lower than those of patients with higher SPA. In addition, Paiva et al. [44] also found that the survival of cancer patients with SPA < −1.65 was on average 2 years less than that of cancer patients with SPA > −1.65 (*p* < 0.001).

The relationship between SPA and survival and the predictive ability of SPA regarding survival have also received increasing attention. Hui et al. [46] reported a positive relationship between SPA and OS in 222 cancer patients who received parenteral hydration (*p* < 0.001), and Yates et al. [47] reported a similar result. In addition, Cereda et al. [53] studied the relationship between SPA and 1-year survival and found that SPA < −1.65 was positively related to 1-year survival in a German cohort (HR 2.00; 95% CI, 1.51–2.66; *p* < 0.001) and an Italian cohort after adjustment (HR 0.724; 95% CI, tertiles of predictor index-0.691; *p* < 0.001). Furthermore, SPA was reported as a significant independent prognostic indicator for survival (univariate analysis models: RR 3.12; 95% CI, 2.03–4.79; *p* < 0.001; multivariate analysis models: RR 2.35; 95% CI, 1.41–3.90; *p* = 0.001) [44], 6-month survival (HR 0.567; 95% CI, 0.470–0.683; *p* < 0.0001) [34], 2-year survival (HR 1.97; *p* = 0.043) [49], and 5-year survival (HR 0.66; 95% CI, 0.52–0.84; *p* < 0.001) [44] in cancer patients. The AUCs for SPA prediction of 6-month survival and survival were 0.734 and 0.66, respectively, in an ROC analysis. However, Pena et al. [13] and Paixao et al. [48] found that SPA was not related to survival, and Yates et al. [47] also reported that there was no relationship between SPA and 30-day or 60-day mortality in cancer patients.

### 3.5. Comparison of the Predictive Ability of SPA and PA

Five studies compared the prediction of nutritional status and clinical outcomes by SPA and PA in cancer patients; the results are shown in Table 7. Different studies have yielded different results. Norman et al. [34] reported that SPA adjusted for sex, age, and BMI enhanced the prognostic relationship of PA not only for nutritional status but also for clinical outcomes. Similarly, Paixao et al. [48] found that SPA (univariate analysis: HR 0.82, 95% CI 0.45–1.51, *p* = 0.527; adjusted analysis: HR 0.65, 95% CI 0.32–1.30, *p* = 0.221) was a better predictor of survival than PA (univariate analysis: HR 1.97, 95% CI 0.68–5.72, *p* = 0.216; adjusted analysis: HR 0.56, 95% CI 0.13–2.36, *p* = 0.427) in cancer patients. Roccamatisi et al. [40] also found that SPA (*p* = 0.032) predicted complications better than PA (*p* = 0.661) in cancer patients. However, Axelsson et al. [45] found that PA was a significant indicator for survival in cancer patients (HR 0.47, *p* < 0.001); it had an AUC of 0.73 in the ROC curve, higher than that of SPA (AUC 0.66). These results showed worse prediction of survival by SPA than by PA. Hui et al. [46] also found a stronger relationship between SPA (γ = 0.11; *p* = 0.11) than PA (γ = 0.075; *p* = 0.28) with clinicians’ predictions of survival in cancer patients but a weaker relationship between SPA and nutritional-status-related indicators such as HGS (SPA: γ = 0.15; *p* = 0.03; PA: γ = 0.35; *p* < 0.001), maximal inspiratory pressure (SPA: γ = 0.04; *p* = 0.60; PA: γ = 0.23; *p* = 0.001), serum albumin (SPA: γ = 0.35; *p* = 0.001; PA: γ = 0.37; *p* < 0.001), fat-free mass (SPA: γ = 0.16; *p* = 0.02; PA: γ = 0.29; *p* < 0.001), and fat-free mass index (SPA: γ = 0.22; *p* = 0.001; PA: γ = 0.33; *p* < 0.001). Based on the above, more studies are needed to confirm and compare the predictive abilities of SPA and PA.

## 4. Discussion

Although studies on predicting outcomes in cancer patients based on BIA have been increasing in number, the role of SPA has not been thoroughly studied. This review was designed to assess the usefulness of BIA-derived SPA in determining nutritional status and predicting clinical outcomes in cancer patients, and to perform a comparative analysis of the predictive efficacies of SPA and PA.

When malnutrition occurs, early membrane permeability increases, body fluid flows from intracellular water (ICW) to extracellular water (ECW), ECW/ICW increases and body cell mass decreases, adversely affecting the electrical properties of tissues, and PA is significantly reduced [57]. Well-nourished patients showed higher PA than malnourished patients [58,59,60,61]. A systematic review reported the predictive ability of PA for nutritional status in advanced cancer patients and found that low PA was related to worse nutritional status as assessed by BMI, serum albumin level, transferrin, and fat-free mass [62]. As a standardized form of PA, SPA has a great similarity to PA. Lower SPA indicated poorer nutritional status, especially malnutrition [63]. Therefore, this review sought to determine the role of SPA in assessing nutritional status in cancer patients. We found that SPA was related to nutritional status and that it showed lower values in HNC patients with elevated nutritional risk [51]; PT-SGA score, BMI and albumin level, all of which are recognized as vital assessment indicators of malnutrition, all showed a downward trend when malnutrition occurred [64,65], and these changes led to a decrease in SPA. Thus, SPA might play a role in predicting malnutrition in cancer patients. We found that SPA was positively related to PT-SGA [13,34] and BMI [53] in patients with mixed cancer and to albumin in patients with AL [47] but was negatively related to the degree of weight loss in patients with mixed cancer [53]. SPA was an independent predictor of nutritional status in cancer patients [34]. In patients with mixed cancer, malnutrition assessed by PT-SGA was also well predicted by SPA [34]. Norman et al. [34] reported that SPA’s predictive ability was enhanced and better than that of PA based on assessment and quantification of individual deviations of cancer patients from population average levels for gender, age, and BMI. SPA offers practical advantages over conventional nutrition assessment methods in that it eliminates the need to measure weight and height when assessing nutritional risk.

Inflammation was identified as an important cause of malnutrition in the diagnostic consensus on malnutrition published by ESPEN [55]. Barrea et al. confirmed a negative relationship between PA and CRP, an inflammation-related factor, and reported the importance of PA in the diagnosis of meta-inflammation [66]. Our review also found that SPA is negatively related to levels of CRP and IL-6, two positive indicators of malnutrition [51]. Cancer cachexia arises from malnutrition [67] and is characterized by loss of muscle mass [68], which causes ECW to increase and ICW to decrease; thus, PA decreases [69]. The studies included in this meta-analysis showed that SPA was positively related to HGS [13,53], MAC [13], and MMA [13]. Based on these findings, SPA might reveal some changes in cancer patients’ cachexia. Moreover, changes in impedance patterns (reduction of capacitive reactance and resistance retention) have been found to occur before overt symptoms of cachexia appear, suggesting a change in the electrical properties of tissues, especially somatic cell mass [70], and SPA decreased. Although patients with severe malnutrition are usually easily identified after screening, dietary assessment, or bedside examination, SPA offers a distinct advantage over current measurement methods for identifying patients without significant malnutrition; thus, SPA appears to be a good predictor of nutritional status in cancer patients.

Assessing the relationship between SPA and clinical outcome helps define the usefulness of SPA in cancer. A previous systematic review demonstrated that PA and SPA predict postoperative complications in cancer patients and encouraged greater reporting of SPA in future work [35]. This review found that, in cancer patients who were undergoing elective surgery (N = 60) or who had mixed [13] (N = 121) or abdominal cancer [40] (N = 182), SPA was significantly negatively related to and even the only predictor of complications. This might be related to the fact that during the occurrence and development of cancer, tumour-derived inflammatory cytokines are released, and homeostasis is damaged, increasing the risk of postoperative infectious complications and affecting the integrity of the cell membrane and somatic cell quality [13], and thus leading to reduced PA and SPA. However, one included study of colorectal cancer patients reported no significant relationship (N = 84) [50]. This might be due to heterogeneity among cancer patients. In addition, SPA and PA have been compared for their accuracy in predicting complications in cancer patients, and SPA was found to have better predictive ability [40]. Similarly, a meta-analysis reported that it was difficult to predict complications using PA in cancer patients due to differences in unadjusted factors such as age and sex, which could influence the interpretation of PA [35]. Therefore, after adjustment for confounders, SPA might be more effective in predicting complications in cancer patients.

Cancer patients often experience metabolic disorders and are prone to malnutrition or cachexia and other conditions that destroy the integrity of cell membranes and cause PA to decrease [71]. Currently, the work of several scholars has led to the use of methods that improve the survival of cancer patients by improving their metabolic status [72], and it was suggested that SPA might be of great significance in assessing survival. The investigation of the relationship and predictive ability of SPA regarding survival in cancer patients presented in this review showed that SPA was positively related to survival in patients with AL [47], patients with advanced cancer who received parenteral hydration treatment [43], and patients with mixed cancer [53]. SPA was also found to be a significant predictor of survival [34,44,45,49]. SPA was a good predictor of 6-month survival in patients with mixed cancers, and its predictive ability was improved compared to PA stratified by sex, age, and BMI [34]. In patients with haematological malignancies, SPA below the 25th percentile was a significant independent predictor of 2-year survival [49]. A stronger prediction of clinical outcomes by SPA than by PA in cancer patients was reported in the included studies [34,40,46,48]. However, the prediction of SPA for long-term survival in patients with stage 1 and 2 mixed cancer was reduced compared with the prediction for short-term survival in patients with stage 3 and 4 cancer [44]. A similar finding was reported for SPA prediction of 5-year survival in patients with HNC; although SPA was adjusted for PA, its prediction was lower than that of PA [45], possibly because SPA, as a prognostic tool, was very sensitive. It corrected for two important negative indicators, increased age and decreased BMI, and increased accuracy while reducing predictability. A 2021 meta-analysis reported that PA was an independent prognostic indicator of survival in patients with advanced cancer after adjustment for any possible confounding factors by multivariate Cox regression analysis [73]. The authors of that study pointed out that SPA could be influenced by adjusting PA for patients with different ethnicities and that this might cause inaccurate predictions when using SPA [73]. In addition, the prediction would change with the passage of time, leading to a decrease in the prediction of long-term survival. Notably, in two different studies, it was also shown that SPA was not related either to survival in patients with mixed cancer [13] or those who were undergoing radiotherapy [48], or to 30- or 60-day survival in patients with AL who were undergoing chemotherapy [47]. The reason for the different effectiveness of SPA in predicting survival might be that SPA was also influenced by the treatment the patients received, in addition to the tumour itself. Chemotherapy affects cell membrane function, calcium channels, and growth receptors [74]. Similarly, radiation has been shown to damage the integrity of cell membranes and increase their permeability [75]. All of these changes could lead to a reduction in SPA. In addition, as a common means of cancer treatment, radiotherapy and chemotherapy can prolong the survival of patients to a certain extent. These factors made it difficult to explore the relationship between SPA and survival rate.

In the clinical setting, determining an appropriate cut-off value for SPA will enable more accurate assessment based on testing nutritional status and clinical outcomes in cancer patients. A previous meta-analysis reported no uniform critical value for PA due to individual differences [76], and a 2021 meta-analysis indicated that the cut-off values of PA ranged from 4.73° to 6° in cancer patients, with a single cut-off value of PA yet to be determined [61]. Although individual bias was accounted for, there was still no uniform cut-off value for SPA in the assessment of nutritional status and clinical outcomes in cancer patients. The cut-off values for SPA in this review varied from −2.26 to 0.3, and the most commonly used cut-off values were −1.65 [13,40,44,45,48,50,51,52] and the 5th percentile [34,46]. However, some studies used the 25th percentile because the Akaike information criterion (AIC) at the 25th percentile (AIC = 529.37) was lower than that at -1.65 (AIC = 531.48) and the 5th percentile (AIC = 530.80) [47,49], suggesting that the assessment of SPA at the 25th percentile was more concise and accurate. Technical factors were also one of the reasons for diversification of the SPA values. As there was no unified international standard, measurement differences in BIA arising from the use of equipment produced by different manufacturers [77] and nonspecification of electroneutral contact electrodes [78] affected the effective measurement of SPA.

Some limitations of this study must be acknowledged. The number of studies that assessed SPA as the main outcome was relatively low (there were only two such multi-centre studies). Due to the different types of BIA instruments used, there was no uniform standard for assessing SPA. All of these factors increased the risk of bias.

## 5. Conclusions

In conclusion, SPA has become an effective objective indicator for assessing the nutritional status of cancer patients. A lower SPA indicates a higher risk of malnutrition and provides a reliable basis on which more appropriate diagnostic and treatment methods can be chosen for clinical cancer patients. However, we still cannot draw definite conclusions about the ability of SPA to predict clinical outcome (including complications and survival rate) in cancer patients. No standard SPA cut-off value has been established. At present, hard evidence is still lacking but, given some promising research results, this may have been caused by the small amount of literature we included. Therefore, additional high-quality studies are needed in the future, and more studies are necessary to clearly prove the relationship between SPA and clinical outcomes of cancer patients, and to establish the critical value of SPA. This will be of great significance in clinical diagnosis, management of treatment, nursing work, and the work of other health professionals in the future.

## Figures and Tables

**Figure 1 nutrients-15-00050-f001:**
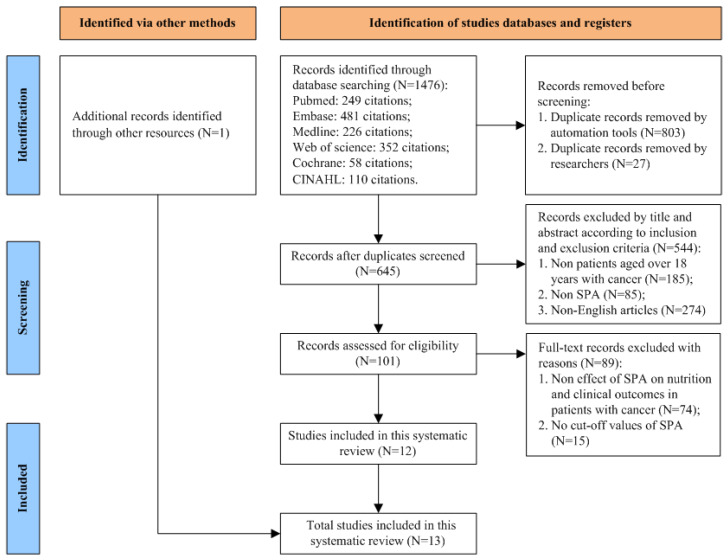
Flow diagram for study selection.

**Figure 2 nutrients-15-00050-f002:**
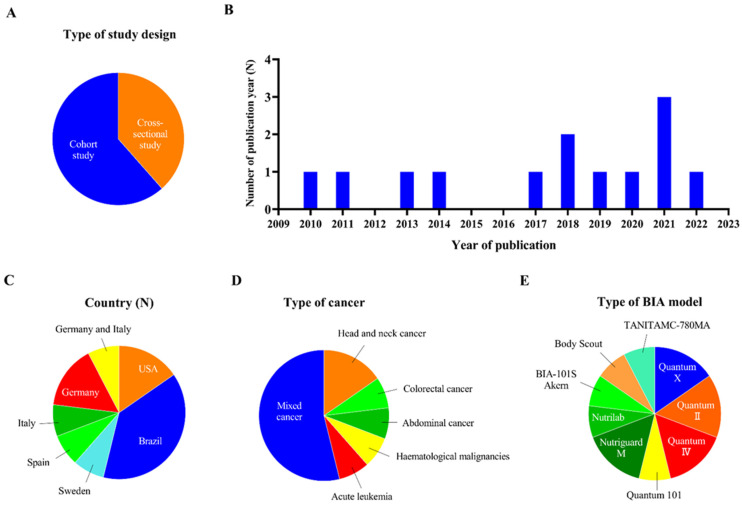
Flow diagram for study selection. (**A**) Pie chart for type of study design of included studies. (**B**) Bar graph for numbers of publication years of included studies. (**C**) Pie chart for countries of included studies. (**D**) Pie chart for types of cancer of included studies. (**E**) Pie chart for types of BIA model of included studies.

**Table 1 nutrients-15-00050-t001:** Search strategy in electronic databases.

Step	Search Strategy
#1	“Phase Angle” OR “Standardized Phase Angle”
#2	Neoplasms OR Neoplasia OR Neoplasias OR Neoplasm OR Tumors OR Tumor OR Cancer OR Cancers OR Malignancy OR Malignancies OR “Malignant Neoplasms” OR “Malignant Neoplasm” OR “Neoplasm, Malignant” OR “Neoplasms, Malignant” OR “Benign Neoplasms” OR “Neoplasms, Benign” OR “Benign Neoplasm” OR “Neoplasm, Benign”
#3	#1 AND #2

**Table 2 nutrients-15-00050-t002:** Characteristics of included cancer patients.

First Author/ Year	Country	Cancer Location/ Sample Size	Gender [Male(%)]	Age (Years)	BMI [Kg/m^2^(%)]	Treatment Methods
Pena, 2019 [13]	Brazil	Mixed, 121	52.9%	58.8 ± 12.5	/ ^3^	Surgical
Axelsson, 2018 [45]	Sweden	HNC ^2^, 128	68%	61.4 ± 11.2	24.9 ± 3.8	Surgery, chemoradiation
Hui, 2014 [46]	USA	Mixed, 222	41%	55	/ ^3^	Parenteral hydration
Yates, 2020 [47]	USA	AL ^1^, 100	44%	59 ± 14.6	29.7 ± 7.1	Intensive induction chemotherapy
Paiva, 2011 [44]	Brazil	Mixed, 195	38%	58 ± 12.9	26.5 ± 5.1	First chemotherapy
Paixao, 2021 [48]	Brazil	Mixed, 62	39%	54.5	25.3	Radiotherapy
Urbain, 2013 [49]	Germany	Haematological Malignancies, 105	62.9%	56.1 ± 10.9	25.9 ± 4.1	Allogeneic hematopoietic cell transplantation
Norman, 2010 [34]	Germany	Mixed, 399	52.1%	63.0 ± 11.8	24.9 ± 4.8	/ ^3^
Maurício, 2018 [50]	Brazil	Colorectal, 84	46.4%	61.6 ± 13.1	/ ^3^	Surgery, neoadjuvant
Leon-Idougourram, 2022 [51]	Spain	HNC ^2^, 45	37.8%	64.5	/ ^3^	Surgery, radiotherapy, chemotherapy
Harter, 2017 [52]	Brazil	Mixed, 60	56.7%	18–39 (18.3), 40–59 (36.7), ≥ 60 (45)	<18.5 (1.7), 18.5–24.9 (30), 25–29.9 (43.3), ≥30 (25)	Elective surgery
Roccamatisi, 2021 [40]	Italy	Abdominal, 182	57.7%	67 ± 11	24.9 ± 4.2	Scheduled to undergo surgical
Cereda, 2021 [53]	Italy, Germany	Mixed, 1084	61.7% 54.8%	64.8 ± 11.6 61.7 ± 12.2	23.3 ± 4.4 24.4 ± 4.1	/ ^3^

Abbreviations: ^1^ AL: acute leukaemia; ^2^ HNC: head and neck cancer; ^3^/: not mentioned.

**Table 3 nutrients-15-00050-t003:** Characteristics of included cancer patients.

First Author/ Year	Model of BIA ^1^	Time of Assessment	Time of Follow-Up	Cut-Off Value of PA	Cut-Off Value of SPA ^2^	Reference Population	Study Design
Pena, 2019 [13]	Quantum X; RJL Systems, Clinton, MI	1 day before surgery	From 1 day after surgery to discharge or death	/ ^3^	−1.65°	Brazilian	Cohort
Axelsson, 2018 [45]	BIA-101S Akern; RJL Systems, Detroit, MI, USA	Time of diagnosis	As long as possible	5.95°	−1.65°	German	Cross- sectional
Hui, 2014 [46]	Quantum IV; RJL Systems, Clinton Township, Mich	Time of admission	Median: 118 days	4.4°	5th	/ ^3^	Cross- sectional
Yates, 2020 [47]	Quantum IV; RJL Systems	Time of diagnosis	60 days	/ ^3^	−0.948°	/ ^3^	Cross- sectional
Paiva, 2011 [44]	Quantum 101; RJL Systems	Before first chemotherapy	3 years and 2 months	/ ^3^	−1.65°	Brazilian	Cross- sectional
Paixao, 2021 [48]	Quantum II; RJL Systems	Before first RT	10 years	/ ^3^	−1.65°	/ ^3^	Cross- sectional
Urbain, 2013 [49]	Body Scout, Fresenius Medical Care, Germany	/ ^3^	2 years	5.06°	25th: −2.26°	German	Cohort
Norman, 2010 [34]	Nutriguard M; Data Input GmbH, Darmstadt, Germany	Within 48 h of admission	6 months	5th	5th	/ ^3^	Cohort
Maurício, 2018 [50]	Quantum X; RJL Systems, Michigan, USA	1 day before surgery	/ ^3^	/ ^3^	−1.65°	Brazilian	Cohort
Leon-Idougourram, 2022 [51]	TANITA MC-780 MA	/ ^3^	/ ^3^	/ ^3^	−1.65°	/ ^3^	Cohort
Harter, 2017 [52]	Quantum II; RJL Systems	Within 48 h after admission	/ ^3^	/ ^3^	−1.65°	/ ^3^	Cohort
Roccamatisi, 2021 [40]	Nutrilab; Akern, Florence, Italy	At 08:00 on the day before scheduled surgery	Within 30 d after discharge	5°	0.3°	/ ^3^	Cohort
Cereda, 2021 [53]	Nutriguard M; data Input Gmbh, Darmstadt Germany	Italian: diagnosis German: different stages of cancer	1 year	/ ^3^	−1.65°	/ ^3^	Cohort

Abbreviations: ^1^ BIA: bioelectrical impedance analysis; ^2^ SPA: standardized phase angle; ^3^/: not mentioned.

**Table 4 nutrients-15-00050-t004:** Relationship between SPA and nutritional status.

Study	Cut-Off Value	Nutritional Indicators Related to SPA ^9^	Main Findings
Pena, 2019 [13]	−1.65°	PT-SGA ^7^, HGS ^3^, MAC ^5^, MMA ^6^	Patients with SPA ^9^ *<* −1.65 had greater chance of malnourishment with low PT-SGA ^7^, MAC ^5^, MMA ^6^, and HGS ^3^.
Yates, 2020 [47]	25th: −0.948°	Albumin	SPA ^9^ < −0.948 was positively related to albumin.
Norman, 2010 [34]	5th	PT-SGA ^7^, EORTC ^2^	SPA ^9^ below 5th percentile value emerged as a significant predictor for malnutrition and impaired functional status in generalized linear model regression analyses.
Leon-Idougourram, 2022 [51]	−1.65°	Arm circumference, calf circumference, BMI ^1^, CRP ^8^, IL-6 ^4^, thigh adipose tissue	Serum CRP ^8^ and IL-6 ^4^ were most reliable parameters for determining patients with decreased SPA ^9^.
Cereda, 2021 [53]	−1.65°	HGS ^3^, BMI ^1^, weight loss	In patients with SPA ^9^ < −1.65, worse nutritional and functional status were observed.

Abbreviations: ^1^ BMI: body mass index; ^2^ EORTC: European Organization for Research and Treatment of Cancer; ^3^ HGS: handgrip strength; ^4^ IL-6: interleucine-6; ^5^ MAC: midarm circumference; ^6^ MMA: muscular midarm circumference; ^7^ PT-SGA: Patient-Generated Subjective Global Assessment; ^8^ CRP: C-reactive protein; ^9^ SPA: standardized phase angle.

**Table 5 nutrients-15-00050-t005:** Relationship between SPA and complications.

Study	Cut-Off Value	Type of Complications	Definition of Complications	Main Findings
Pena, 2019 [13]	−1.65°	Infectious complications; Non-infectious complications	Bulletin of the American College of Surgeons	Patients with SPA^1^ *<* −1.65 presented more infectious complications, but there was no association between SPA ^1^ and other complications. SPA ^1^ was only one significant predictor of infectious complications.
Harter, 2017 [52]	−1.65°	Postoperative complications	Clavien–Dindo classification	SPA ^1^ was significantly lower among those who had severe postoperative complications.
Maurício, 2018 [50]	−1.65°	Postoperative complications	Clavien–Dindo classification	SPA ^1^ showed no association with postoperative complications in cancer patients.
Roccamatisi, 2021 [40]	0.3°	Infectious complications	Clavien–Dindo classification	SPA ^1^ was significantly lower in patients with infectious complications. SPA ^1^ < 0.3 was the only independent variable for infectious complications.

Abbreviations: ^1^ SPA: standardized phase angle.

**Table 6 nutrients-15-00050-t006:** Relationship between SPA and survival.

Study	Cut-Off Value	Type of Survival	Main Findings
Pena, 2019 [13]	−1.65°	Survival	There was no association between SPA ^3^ and survival.
Urbain, 2013 [49]	25th: −2.26°	2-year survival	SPA ^3^ < −2.26 was a significant independent predictor for 2-year survival in cancer patients.
Axelsson, 2018 [45]	−1.65°	5-year survival	SPA ^3^ < −1.65 was a significant prognostic indicator for 5-year survival in cancer patients.
Hui, 2014 [46]	5th	OS ^1^	SPA ^3^ below 5th percentile value was found to be significantly related to OS ^1^.
Yates, 2020 [47]	−0.948°	OS ^1^, 30-day mortality, 60-day mortality	SPA ^3^ < −0.948 was positively related to OS ^1^, while there was no relationship between SPA ^3^ and 30-day or 60-day mortality in cancer patients.
Paiva, 2011 [44]	−1.65°	Survival	SPA ^3^ < −1.65 was a significant determining indicator of higher mortality in cancer patients.
Paixao, 2021 [48]	−1.65°	Survival	SPA ^3^ was not related to survival in cancer patients during RT ^2^.
Norman, 2010 [34]	5th	6-month survival	SPA ^3^ below 5th percentile value was an independent predictor for 6-month mortality of cancer patients.
Cereda, 2021 [53]	−1.65°	1-year survival	SPA ^3^ < −1.65 was positively related to 1-year survival of German cohort and Italian cohort after adjusting in cancer patients.

Abbreviations: ^1^ OS: overall survival; ^2^ RT: radiotherapy; ^3^ SPA: standardized phase angle.

**Table 7 nutrients-15-00050-t007:** Comparison of the Predictive Ability of SPA and PA.

Study	Cut-Off Value of PA ^1^	Cut-Off Value of SPA ^2^	*p* Value or AUC ^3^ of PA ^1^	*p* Value or AUC ^3^ of SPA ^2^	Comparison
**Nutrition** **al** **status**					
Norman, 2010 [34]	5th	5th	/	/	SPA ^2^ > PA ^1^
Hui, 2014 [46]	4.4°	5th	HGS: *p* < 0.001; Maximal inspiratory pressure: *p* = 0.001; serum albumin: *p* < 0.001; fat-free mass: *p* < 0.001; fat-free mass index: *p* < 0.001.	HGS: *p* = 0.03; Maximal inspiratory pressure: *p* = 0.60; serum albumin: *p* = 0.001; fat-free mass: *p* = 0.02; fat-free mass index: *p* = 0.001.	SPA ^2^ < PA ^1^
**Clinical outcomes**					
Complications					
Roccamatisi, 2021 [40]	5°	0.3°	*p* = 0.661	*p* = 0.032	SPA ^2^ > PA ^1^
Survival					
Norman, 2010 [34]	5th	5th	/	/	SPA ^2^ > PA ^1^
Paixao, 2021 [48]	/	−1.65°	Univariate analysis: *p* = 0.216;Adjusted analysis: *p* = 0.427	Univariate analysis: *p* =0.527; Adjusted analysis: *p* = 0.221	SPA ^2^ > PA ^1^
Axelsson, 2018 [45]	5.95°	−1.65°	AUC ^3^ = 0.73	AUC ^3^ = 0.66	SPA ^2^ < PA ^1^
Hui, 2014 [46]	4.4°	5th	*p* = 0.28	*p* = 0.11	SPA ^2^ > PA ^1^

Abbreviations: ^1^ PA: phase angle; ^2^ SPA: standardized phase angle; ^3^ AUC: area under the curve.

## Data Availability

Not applicable.

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
