# Peer review of "The Role of Standardized Phase Angle in the Assessment of Nutritional Status and Clinical Outcomes in Cancer Patients: A Systematic Review of the Literature"

_nutrients, 2022, doi:10.3390/nu15010050_

Round 1

Reviewer 1 Report

Dear authors,

a very significant paper. Improving or maintaining body weight during cancer treatment is known to be of significant impact on survival and diminishing (severity of) side effects. 

Your manuscript however was very difficult to read because the English is not up to standard. There are missing words, there are words that make no sense. The context of the sentences is therefore difficult to comprehend. 

I would suggest a thorough review by a native English speaker and than resubmit. 

Yours sincerely,

the reviewer

Author Response

请参阅附件。

Reviewer 2 Report

I suggest some suggestions and comments:

Line 15: which databases? should also be in the abstract

Line 15: including 2787 participants should be here the Eligibility Criteria: The included studies considered the following criteria

Line 80: should be “Risk of study bias” ex. MOOLA, SZCEKRE, et al. Chapter 7: Systematic reviews of etiology and risk. Joanna Briggs Institute Reviewer's Manual. The Joanna Briggs Institute, 2017, 5.

Line 95: Exclusion criteria: 1) Participants were not patients aged ≥18 should replace with ≤ 18

Line 169: low levels of albumin suggested a high risk of malnutrition 169 Albumin is not a specific parameter of malnutrition and is altered in several pathologies, but pre-albumin is more specific for malnutrition

Line 212: The parenthesis is missing ( and the p is not italicized

Line 335: And work of health professionals in the future

Round 2

Reviewer 1 Report

Dear authors,

thank you very much for adjusting the English of the paper. As expected this has created an easy to read paper, which adds to the literature regarding the importance of malnutrition in cancer patients and their survival and quality of life. Even though the clear evidence is still lacking, as shown indications are there and warrant further study. This papers again indicates that. 

Except, I presume a "c" missing in achexia, line 351 page 13. I do not have any suggestions for adjusment. 

Thank you very much for your investigations and revision in this important field of oncology. Looking forward to research adding to the evidence.  

With kind regards,

Reviewer.